# Majorana bound states in encapsulated bilayer graphene

Fernando Peñaranda,[1] Ramón Aguado,[1] Elsa Prada,[1] and Pablo San-Jose[1]

[1]*Instituto de Ciencia de Materiales de Madrid, Consejo Superior de Investigaciones Científicas (ICMM-CSIC), Madrid, Spain*

(Dated: April 21, 2023)

The search for robust topological superconductivity and Majorana bound states continues, exploring both one-dimensional (1D) systems such as semiconducting nanowires and two-dimensional (2D) platforms. In this work we study a 2D approach based on graphene bilayers encapsulated in transition metal dichalcogenides that, unlike previous proposals involving the Quantum Hall regime in graphene, requires weaker magnetic fields and does not rely on interactions. The encapsulation induces strong spin-orbit coupling on the graphene bilayer, which opens a sizeable gap and stabilizes fragile pairs of helical edge states. We show that, when subject to an in-plane Zeeman field, armchair edges can be transformed into p-wave one-dimensional topological superconductors by contacting them laterally with conventional superconductors. We demonstrate the emergence of Majorana bound states (MBSs) at the sample corners of crystallographically perfect flakes, belonging either to the D or the BDI symmetry classes depending on parameters. We compute the phase diagram, the resilience of MBSs against imperfections, and their manifestation as a $4\pi$-periodic effect in Josephson junction geometries, all suggesting the existence of a topological phase within experimental reach.

Majorana bound states (MBSs) were predicted by Kitaev in 2001[1] as the fractionalized, zero-energy, protected fermion states that develop at the boundaries of one-dimensional (1D) topological superconductors. Interest in these states quickly grew past fundamental research, as it was realized that their spatial wavefunction non-locality could enable, in principle, scalable protection of quantum information[2–7]. A practical proposal to engineer MBSs in proximitized Rashba nanowires was made by Oreg. *et al.* and Lutchyn *et al.* a few years later[8,9], soon followed by the first experiments[10], which revealed promising hints of potential MBSs. Since these hallmark results the story of MBSs in nanowires has grown increasingly complex[11,12]. Remarkable fabrication improvements and careful experimental characterization[13,14] have now clearly confirmed the existence of zero modes in these systems[15], but have also revealed significant interpretation issues and departures from theoretical expectation in their behavior[11]. The reasons are varied, and are thought to include disorder[16,17], electrostatics[18,19], metallization[20] and non-topological near-zero energy states due to confinement effects, including quantum dot formation[21,22] and smooth potentials[23–26]. One decade after their theoretical proposal, proximitized nanowires have evolved into the most studied and advanced solid state platform for topological superconductivity. However, we have still not been able to conclusively demonstrate the predicted topological MBSs, let alone harness their potential for quantum computation.

This state of affairs has pushed numerous researchers to explore alternative experimental platforms for topological superconductivity (TSC), including atomic chains[27,28], 2D semiconducting heterostructures[29,30], planar Josephson junctions[31,32], full-shell nanowires[22,33], graphene-based platforms[34,35], several 2D crystals[36–38] and van der Waals heterostructures[39]. Many of the proposals for 1D TSCs start from the basic Fu-Kane recipe[40,41]: couple an s-wave superconductor to a 1D

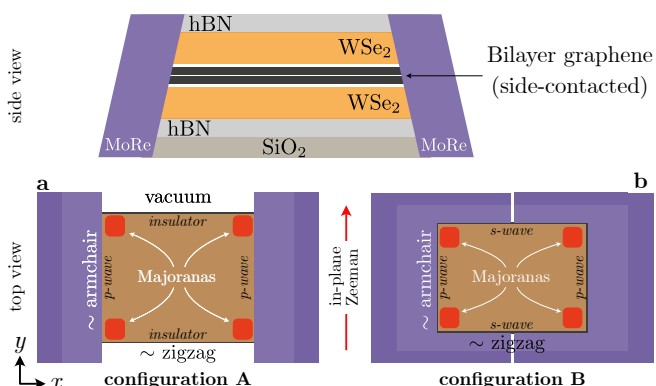

FIG. 1. Lateral and top views of proposed device configurations A and B for the generation of Majorana bound states (MBSs) (schematically represented in red). The device is composed of a graphene bilayer (black), encapsulated in a transition metal dichalcogenide (TMDC) (orange) and laterally contacted with conventional s-wave superconductors (purple). The superconductor split in (b) creates a weak link that allows to phase-bias the junction.

spinless electron liquid with finite helicity (i.e. to non-degenerate 1D modes with some degree of spin-momentum locking, such as the edge states of a 2D Quantum Spin Hall system[42,43]). The s-wave pairing opens a finite p-wave TSC gap on the helical liquid[4], and gives rise to zero-energy MBSs at boundaries with trivial gaps. The various implementations of this recipe typically differ in the mechanism that generates the spinless helical phase. For example, in the original 1D Rashba nanowires proposal[8,9] it is a combination of Rashba spin-orbit coupling (SOC), Zeeman field and low electron densities.

We focus here on graphene-based approaches to MBSs. Graphene allows for exquisitely clean electronics[44,45] and good superconducting proximity effect under magnetic fields[46–49], properties that could help overcome some of the material-specific problems of Majorana nanowires.

In Ref. 50 it was experimentally demonstrated that the $\nu = 0$ quantum Hall state of monolayer graphene behaves, under a strong in-plane magnetic field, as a quantum spin Hall state, an observation explained as the result of Zeeman polarization of an antiferromagnetic ground state induced by strong electron-electron interactions[51,52]. In Ref. 34 it was shown that a 1D TSC could be created on such a polarized $\nu = 0$ quantum Hall state by proximitizing its edges. The proposed configuration, while conceptually correct for the purpose of generating Majoranas, was experimentally problematic, since s-wave pairing breaks down quickly under the required magnetic fields, thus making the proximity effect of the $\nu = 0$ state challenging. A subsequent proposal was put forward that does away with the need of strong in-plane magnetic fields by employing twisted bilayer graphene in the QH regime under a strong perpendicular electric field[35,53]. The latter is used to transform the bilayer QH edge states into a spinless helical phase by tuning each layer to an opposite filling factor $\nu = \pm 1$. This proposal, however, still requires strong electron-electron interactions to trigger the helical spin structure, as it exploits the ferrimagnetic sublattice polarization induced by interactions to stabilize $\nu = \pm 1$ QH plateaus. Furthermore, it presents other potential problems such as a reduced topological gap and a required electron-hole character of the bilayer, which could hinder superconducting pairing by a nearby superconductor. Other proposed avenues towards MBSs based on electronic interactions include the use of intrinsic superconducting correlations in magic-angle twisted graphene bilayers in combination with other 2D crystals[54].

In this work we present a third kind of approach to MBSs in graphene that does not rely on the QH effect or electron-electron interactions. Instead, it exploits the strong spin-orbit coupling (SOC) induced onto a Bernal-stacked graphene bilayer when it is encapsulated in a semiconducting transition metal dichalcogenide (TMDC) such as $WSe_2$, see Fig. 1(top). The SOC gaps the bulk of the bilayer[55–60], and is thought to be responsible for anomalies observed in different graphene-based Josephson junctions[61,62]. In the bilayer, the induced SOC produces Kramers pairs of counterpropagating topologically fragile edge states at the boundaries, see red and blue lines in Fig. 2(a,b). We show that some of these boundaries can develop spinless helical 1D modes under small Zeeman fields. We use here the term spinless helicity, as is conventional, to denote the existence of an odd number of pairs of non-degenerate counterpropagating modes at a given energy and edge, whose spin depends on the direction of propagation. The development of spinless helicity depends on edge crystallographic orientation. It is optimal for armchair edges and is absent for zigzag edges. A spinless helical edge can be gapped into a p-wave superconductor by side-contacting it to an s-wave superconductor[40]. MBSs then arise at the corners of the sample (see Fig. 1) above a Zeeman field comparable to the induced superconducting gap, as in Majorana nanowires.

Despite their dependence on the crystallographic orientation of the edges, we show that MBSs are resilient to a certain amount of contact disorder and misalignment, and exhibit the expected $4\pi$-periodic topological Josephson effect[41,63]. Our analysis also reveals the appearance of an intriguing regime with pairs of near-zero modes at each corner, analogous to the approximate BDI-class MBSs of narrow multimode nanowires[64,65], that occupies a large portion of parameter space around charge neutrality.

## I. EDGE MODES IN ENCAPSULATED BILAYER GRAPHENE

TMDCs are semiconducting 2D crystals, such as $WSe_2$ or $MoS_2$, with strong spin-orbit. The possibility of inducing a strong SOC on graphene monolayers by placing it in contact to a TMDC was demonstrated using a variety of theoretical[66–68] and experimental techniques[55–60]. Two main types of SOC are generated on the low-energy sector of monolayer graphene close to the neutrality point: Ising and Rashba[59,60,67]. At low energies these two couplings can be written as

$$H_I = \frac{\lambda_I}{2}\tau_z s_z, \tag{1}$$

$$H_R = \frac{\lambda_R}{2}(\sigma_x \tau_z s_y - \sigma_y s_x). \tag{2}$$

in terms of the valley ($\boldsymbol{\tau}$), spin ($\boldsymbol{s}$) and pseudospin ($\boldsymbol{\sigma}$) Pauli matrices, which act on the subspace of the $K$ and $K'$ valleys, the physical electron spin and the carbon sublattices within the graphene unit cell, respectively.

The expected magnitude of the couplings is rather sizable, of the order of $\lambda_R \lesssim \lambda_I \approx 2 - 3$ meV in the case of $WSe_2$[59] (possibly larger for $WS_2$[55,56]) and depends strongly on the interlayer rotation angle with graphene[66]. The low-energy model for graphene becomes $H = H_0 + H_I + H_R$, where $H_0 = v_F(\tau_z k_x \sigma_x + k_y \sigma_y)$ is the Dirac Hamiltonian for an isolated graphene monolayer and $v_F$ is the Fermi velocity.

In the case of a graphene bilayer encapsulated on both sides with lattice-aligned $WSe_2$, each layer acquires the above couplings, with the peculiarity that the corresponding $\lambda_I$ and $\lambda_R$ have an opposite sign on each layer[59,69]. In the low-energy sector of bilayer graphene the pseudospin is equal to the layer quantum number[70], so that the low-energy effective model for bilayer graphene with a simple Bernal interlayer hopping $t_1$ (i.e. neglecting trigonal warping[70]) becomes

$$H = \frac{v_F k^2}{t_1}(\tau_z \sigma_x k_x + \sigma_y k_y)^2 + \frac{\tilde{\lambda}_I}{2}\tau_z s_z \sigma_z + \mathcal{O}(k^3),$$

$$\tilde{\lambda}_I = \left(1 - 2\frac{v_F^2 k^2}{t_1^2}\right)\lambda_I. \tag{3}$$

Note that $H_R$ does not contribute to the low-energy bulk modes to this order. The $H_I$, in contrast, becomes a

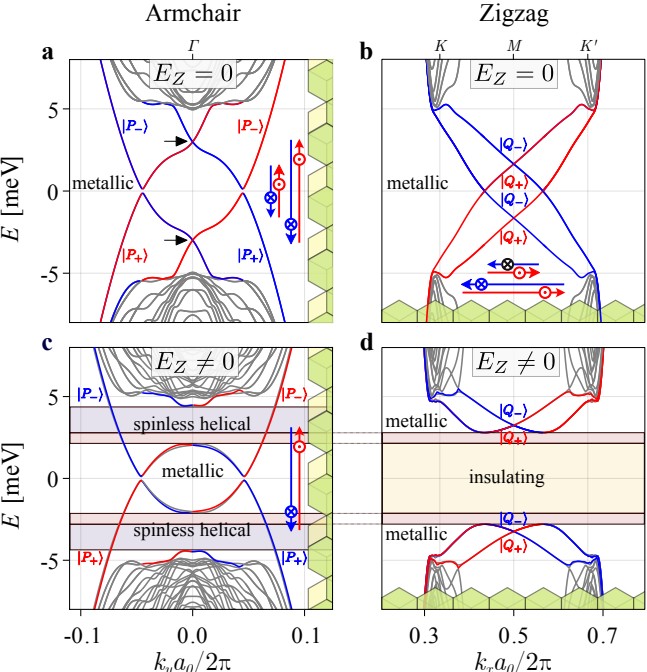

FIG. 2. Dispersion and spin structure of edge modes along armchair (a,c) and zigzag (b,d) edges of a graphene bilayer flake where a full TMDC encapsulation opens a gap $\lambda_I = 10$ meV. Red and blue denote subgap modes propagating along a given edge with opposite out of plane spin polarization ($\odot$ and $\otimes$), which is locked to momentum as shown in the insets. $|P_\pm\rangle$ and $|Q_\pm\rangle$ denote an additional quantum numbers due to orbital symmetries $P$ and $Q$, see text. On the bottom row we show the effect of an in-plane Zeeman field $E_Z$ on the edge modes. On an armchair edge (c) $E_Z$ opens helical windows around the $k_y = 0$ ($\Gamma$-point) band crossings [black arrows in (a)], while on a zigzag edge (d) it opens an insulating gap around zero energy. The different combinations of armchair/zigzag phases are encoded in each energy interval by a white, purple, salmon and yellow background (see also Fig. 3).

Kane-Mele coupling[42], which in the monolayer would open a topological QSH gap at the Dirac point of magnitude $\sim \lambda_I$. Here, $\lambda_I$ is much larger than the (impractically small) intrinsic Kane-Mele term of the monolayer, but is expected to open a topologically trivial gap due to the $2\pi$ Berry phase of each valley in the bilayer (as opposed to $\pi$ in the monolayer) with pairs of topologically fragile helical modes on each edge inside it[69]. We confirm this expectation below. Despite their technically fragile nature, we note that the helical edge states are robust against a wide range of disorder, in particular any form of spin-independent disorder on the lattice, including vacancies or other valley-mixing perturbation (see App. A). The reason is that, as will be shown promptly, their helicity is exact, in the sense that counterpropagating edge states have opposite out-of-plane spin $s_z$ on any edge, so backscattering requires a spin-active perturbation.

To understand the structure of SOC-induced edge

states we numerically simulate the bandstructure of graphene bilayer nanoribbons with both armchair and zigzag edges. The bilayer is modeled with a Bernal-stacked tight-binding Hamiltonian[70]. To reach experimental sizes (particularly important in the next section) we use a scaled lattice constant, with hopping parameters also scaled to keep low-energy observables scaling-independent[71]. On each layer we add the SOC terms $H_I$ and $H_R$ with opposite sign. The resulting bandstructures are show in Fig. 2 for armchair nanoribbons (left column) and zigzag nanoribbons (right column).

The effective low-energy Kane-Mele coupling is indeed found to open a SOC gap, with two pairs of counterpropagating states on each edge. Spin-symmetry is broken, with two distinct propagating modes of opposite spin out-of-plane for each edge and propagation direction. These states are shown in Fig. 2(a,b), with red and blue denoting their spin orientation. Despite the fact that Rashba SOC $H_R$ does not enter the low-energy Hamiltonian of bulk modes, it does affect the edge modes. For armchair edges, in particular, it constitutes a weak, time-reversal-symmetric, gap-opening perturbation around zero energy (charge neutrality point), see Fig. 2(a).

If we neglect Rashba, we find that armchair edge states $|k_y\rangle$ have a second (orbital) quantum number, independent of the spin and associated to their behavior under the parity operator $P = \sigma_x \mathcal{K}$, where $\mathcal{K}$ is conjugation and $\sigma_x$ exchanges layers and sublattices. This quantum number is $\eta = \langle -k_y | P | k_y \rangle = \pm 1$, and its value for each mode is indicated by $|P_+\rangle$ (even) and $|P_-\rangle$ (odd) in Figs. 2(a,c). In the zigzag case all subbands are even under parity, but at the $M$-point crossings ($k_x a_0 / 2\pi = 0.5$ in Fig. 2), edge states $|M\rangle$ can be classified by a second orbital symmetry $Q = \sigma_y$, where $\sigma_y$ is now defined to act on the two columns of sites in the unit cell perpendicular to the edge. Unlike $P$, the $Q$ symmetry is just approximate, but quickly becomes exact in the limit of small $a_0$. The corresponding quantum number $\eta' = \langle M | Q | M \rangle = \pm 1$ of each band is denoted in Figs. 2(b,d) as $|Q_\pm\rangle$. These orbital symmetries are important to understand the splitting of the edge modes under an in-plane Zeeman field.

Let us focus first on the $\Gamma$-point crossing at finite energy in the armchair edge states, see the black arrows in Fig. 2(a). Both of these are crossings between states of equal parity $\eta$. The addition of a Zeeman field along the $y$ direction

$$H_Z = E_Z \sigma_y \qquad (4)$$

preserves parity but breaks the time-reversal symmetry, and immediately turns the crossings into anticrossings. This is illustrated in Fig. 2(c). The reason is the opposite (helical) out-of-plane spin orientation of the armchair states crossing at $k_y = 0$, see the inset sketch. The out-of-plane spin polarization is due to the dominant Ising SOC $H_I$ of Eq. (1). The in-plane Zeeman $H_Z$ mixes the crossing modes, opening energy windows inside the SOC gap (shaded in purple and salmon color) wherein armchair edges support spinless helical edge modes. In

contrast, for the crossings at $k_y \neq 0$ and zero energy, the crossing modes have opposite parity, which prevents their splitting (unless Rashba is non-zero).

Zigzag edge modes behave in the opposite way, acquiring a full gap around zero energy, while the crossings at the $M$ point remain unsplit owing to the opposite $\eta'$ of the crossing modes. Depending on the value of the chemical potential inside the SOC gap, zigzag edges can therefore be either insulating or metallic (i.e. with spinful edge modes as in the absence of Zeeman field), but never spinless. There are then four distinct combinations possible in a vacuum terminated (normal) sample, corresponding to either spinless helical or metallic armchair edges and to insulating or metallic zigzag edges. We encode these four phases in white, purple, yellow and salmon throughout this work, see Fig. 2. The sample can be tuned to any of the four by adjusting Zeeman and chemical potential. Note that here and in the following, we use the term 'metallic' to denote edges where an even number of counterpropagating edge modes coexist at a given energy, in contrast to the case of a spinless helical edge with an odd number of them.

## II. SUPERCONDUCTING PROXIMITY EFFECT AND MAJORANAS

For the purposes of implementing a Fu-Kane approach to generate MBSs in this system we need to introduce superconducting pairing correlations on the spinless helical edge modes. We follow the conventional route of inducing superconductivity externally by contacting a conventional superconductor laterally to the encapsulated bilayer, a technique that has been extensively demonstrated[46–49]. We analyze two distinct geometries, see Fig 1. Configuration A, Fig. 1(a), has proximitized armchair and vacuum-terminated zigzag edges, while in B, Fig. 1(b), the zigzag edges are also contacted to a superconductor (possibly with a weak link to allow phase-biasing the junction, though this detail can be ignored until Sec. V). The superconducting proximity effect is modeled as a pairing term $\Delta$ on the boundary sites of each edge, although the results are qualitatively similar with a more elaborate model where a square-lattice superconductor is explicitly incorporated in the tight-binding lattice.

We compute their corresponding phase diagrams in each configuration, see Figs. 3(a,b), by locating $\Gamma$-point band inversions in sufficiently wide infinite nanoribbons, with either armchair/superconductor or zigzag/superconductor edges. We find that proximitized edges of any type develop a trivial s-wave gap at zero Zeeman field (white region in the phase diagrams). On proximitized armchair edges tuned to their spinless helical window, a Zeeman energy above the effective induced pairing $\Delta^*_{\mathrm{AC}}$ (here $\approx 0.21$ meV for the chosen value of $\Delta = 0.3$ meV), creates a band inversion into a topological p-wave phase (purple and salmon-colored regions), as

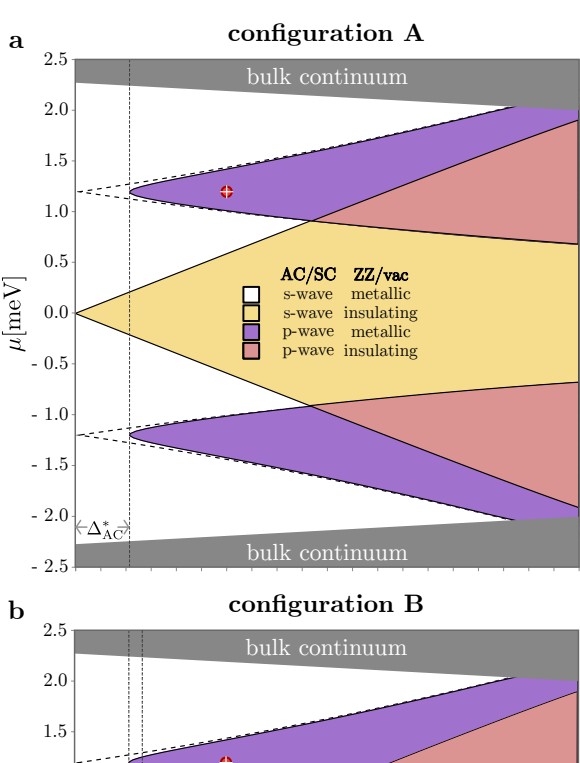

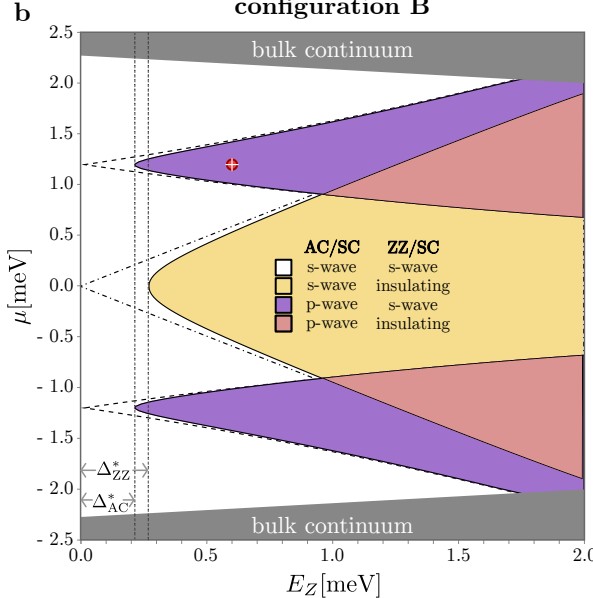

FIG. 3. Phase diagrams of an encapsulated bilayer with induced SOC $\lambda_I = 5$ meV versus Zeeman and chemical potential. Panels (a) and (b) correspond to configurations A and B in Fig. 1, respectively. Each region is defined by different types of edge states along armchair (AC) and zigzag (ZZ) edges, terminated with either vacuum (vac) or a superconductor (SC), see legend for each configuration. An induced pairing $\Delta = 0.3$ meV is applied to any edge sites in direct contact to a SC. Dashed (dash-dotted) lines are metallic/helical (metallic/insulating) boundaries in AC/vac (ZZ/vac) edges. Vertical dotted lines indicate effective induced gaps $\Delta^*$ in AC and ZZ edges. In the salmon-colored regions of both phase diagrams, the system develops a D-class MBS at each sample corner [where a p-wave AC/SC edge and an insulating ZZ/vac (insulating ZZ/SC) meet in configuration A (B)]. In the purple region, corner MBSs also appear in configuration B, whereas Majorana states delocalize along the ZZ/vac metallic edges in configuration A. In the yellow region, BDI-class *pairs* of MBSs develop at each corner for both configurations if Rashba SOC is neglected (see text for details).

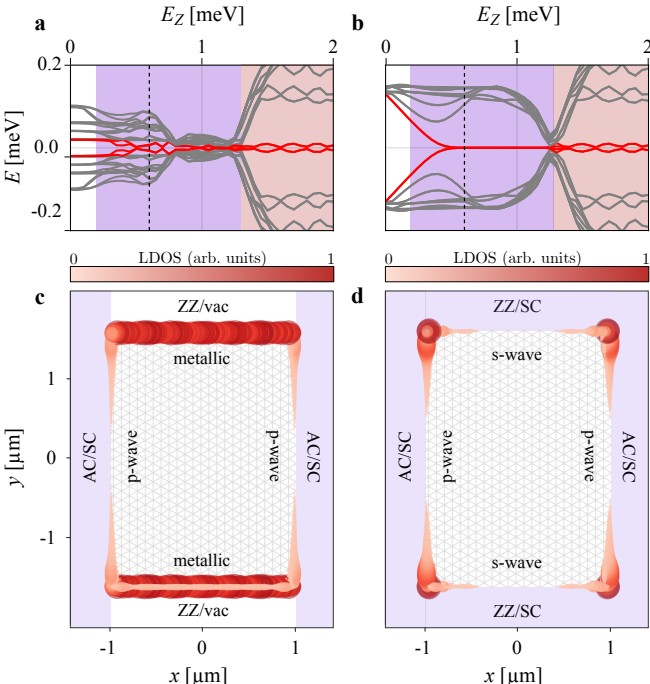

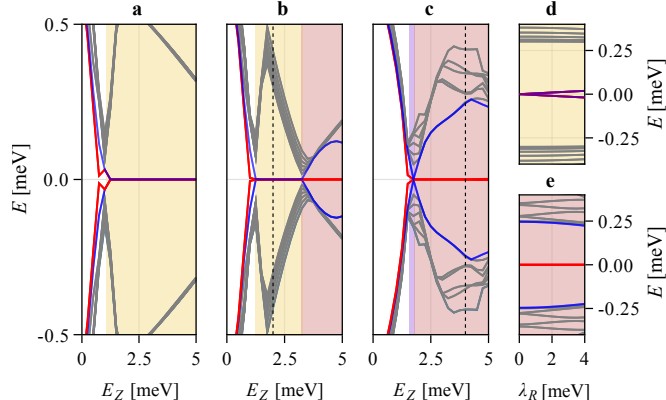

FIG. 4. (a,b) Low-energy spectrum as a function of Zeeman splitting $E_Z$ of a rectangular sample in the two configurations A and B of Fig. 1 and with the same parameters as Fig. 3. Background colors match Fig. 3. (c,d) Local density of states across the sample corresponding to the four lowest eigenstates [red curves in (a,b)] at the vertical dashed line in (a,b) (p-wave armchair phase). In configuration A (a,c), the zigzag edges are metallic, so the MBSs become spatially delocalized and merge into a quasi-continuum of zigzag states, while in configuration B (b,d) zigzag edges have a trivial s-wave gap, so the MBS remain localized at the corners.

FIG. 5. Low-energy spectrum as a function of $E_Z$ of a $2\mu\text{m} \times 2\mu\text{m}$ device similar to Fig. 4d but with $\lambda_R$ set to zero and $\Delta$ increased to 1meV. Panels (a-c) correspond to $\mu = 0$ (a), $\mu = 0.6$meV (b) and $\mu = 1.2$meV (c). Background colors correspond to different regions of the phase diagram, similar to Fig. 3(b) but with phase boundaries pushed to larger $E_Z$ due to the increased $\Delta$. For $E_Z \gtrsim \Delta^*_{ZZ}$, in the yellow regions, unexpected pairs of Majorana zero modes at each sample corner appear that correspond to a BDI-class $\mathbb{Z}$-invariant $\nu^{AC}_{BDI} = 2$ for the AC edge and $\nu^{ZZ}_{BDI} = 0$ for the ZZ one. The zero-energy eigenvalues in the yellow region are therefore eightfold-degenerate. These become near-zero modes as the BDI symmetry is slightly broken by a finite Rashba coupling $\lambda_R$ (d). In the regions with salmon-colored background of (b) and (c) the armchair edge has a $\nu^{AC}_D = \nu^{AC}_{BDI} = 1$ invariant, regardless of symmetry class. Thus, the four zero-energy MBSs (red lines), one at each sample corner, remain insensitive to Rashba (e). Vertical dashed lines in (b,c) correspond to the $E_Z$ used in (d,e), respectively.

predicted by Fu and Kane[4,40]. In contrast, on a proximitized zigzag edge close to neutrality, $\mu = 0$, a Zeeman that exceeds the corresponding $\Delta^*_{ZZ} \approx 0.27$ meV transforms the s-wave phase into an insulator [yellow and salmon-colored regions in Fig. 3(b)]. As a result, a device in configuration A or B within the salmon-colored region (strong Zeeman fields) should localize a MBS at each of its armchair/zigzag corners, as these are boundaries between topological (p-wave) and trivial (insulating) edges. However, within purple regions (weaker Zeeman) only configuration B should host localized corner MBSs (corners become p-wave/s-wave boundaries). Configuration A should instead delocalize its corner MBSs along the metallic zigzag edges.

These predictions are readily confirmed by numerical simulations of large but finite-size rectangular samples in both A and B configurations. In Fig. 4 we show the low-energy eigenvalues for A and B samples (top row) as a function of $E_Z$, and the local density of states (LDOS, bottom row) corresponding to the lowest (red) eigenstates. For both configurations (left and right columns) we choose a point within the purple region (marked with a red dot in Fig. 3). As anticipated, the LDOS exhibits spatially localized/delocalized MBSs in the B/A configurations as described above. The energy of localized MBSs in Fig. 4(d) remains pinned to zero within the purple region, but eventually becomes finite in the salmon-colored region due to finite-size effects (splitting due to MBS overlap). In contrast, delocalized MBSs in configuration A, purple region, strongly hybridize along the zigzag edge with the MBS at the opposite corner, splitting and merging into a quasi-continuous set of finite energy Andreev bound states.

## III. BDI-CLASS MAJORANA PAIRS

To complete the analysis of the phase diagram we now show the spectrum within the yellow regions of Fig. 3(a,b). Focusing on configuration B at zero chemical potential $\mu = 0$, one would expect an s-wave gap along the armchair edge, and an s-wave or insulating gap for $E_Z < \Delta^*_{ZZ}$ and $E_Z > \Delta^*_{ZZ}$, respectively. In both cases, the generic expectation is therefore to have no zero-modes. Surprisingly, however, the spectrum shows a multiply-degenerate *near-zero mode* at $E_Z > \Delta^*_{ZZ}$ (insulating zigzag edge). These states become exact zero

modes when we remove the Rashba coupling, $\lambda_R = 0$. The results are shown in Fig. 5 at $\mu = 0$ (a), $\mu = 0.6$ meV (b) and $\mu = 1.2$ meV (c), and as a function of $\lambda_R$ (d,e), for the same parameters as in Figs. 3 and 4 but for a longer $2\mu$m junction along $x$ and an increased $\Delta = 1$ meV.

To understand the nature of these unexpected zero modes we must recall the phenomenology of multimode Rashba nanowires, which may also exhibit multiple near-zero modes at either end when an even number of modes become topological. When the number of inverted modes is even, the wire is technically in a trivial D-class phase with $\nu_D = 0$ invariant ($\nu_D \in \mathbb{Z}_2$), so no protected zero energy MBSs are expected. It was shown[65,72], however, that a hidden BDI-symmetry[73] emerges if the SOC-induced inter-mode coupling vanishes, which is a good approximation for nanowires of width much smaller than the spin-orbit length. In such limit the nanowire Hamiltonian can be cast into a real matrix belonging to the BDI symmetry class, albeit one where time-reveral symmetry (TRS) $\tilde{\mathcal{T}} = is_y\mathcal{K}$ is broken by Zeeman, and a pseudo-TRS $\tilde{\mathcal{T}} = \mathcal{K}$ (conjugation) takes its place. The BDI invariant in 1D is $\nu_{\mathrm{BDI}} \in \mathbb{Z}$. The total number of zero modes at a nanowire boundary then becomes the difference in $\nu_{\mathrm{BDI}}$ at either side of the boundary, which can be more than one[65,72]. In nanowires the value of $\nu_{\mathrm{BDI}}$ actually matches the total number of spinless modes that have undergone a topological transition. For small but finite $E_Z$, such that no modes have transitioned yet, it is therefore $\nu_{\mathrm{BDI}} = 0$ (like in vacuum). This trivial invariant can also be physically understood as a consequence of the *opposite helicity* of two pairs of modes weakly split by Zeeman in a Rashba nanowire.

A similar situation applies to the armchair edge in our encapsulated bilayer. The low-energy Hamiltonian Eq. (3) for a nanoribbon with proximitized armchair edges and zero Rashba $\lambda_R = 0$ can be cast into a real form, so its symmetry class effectively becomes BDI in this limit, with invariant $\nu_{\mathrm{BDI}}^{\mathrm{AC}} \in \mathbb{Z}$ when TRS is broken by a finite Zeeman field. A crucial difference with nanowires, however, is that the spinful edge modes along a given armchair edge do not have zero net helicity: the two pairs of modes in a given edge have an *equal* (instead of opposite) helicity, determined by the sign of $\lambda_I$ [see inset in Fig. 2(a)]. As a consequence, the $\mathbb{Z}$ BDI invariant in an armchair edge at small $E_Z$ (and actually all throughout the white and yellow regions of Fig. 3) is $\nu_{\mathrm{BDI}}^{\mathrm{AC}} = 2$, not zero. This has the dramatic implication that *pairs* of localized MBSs will arise at each corner as soon as the zigzag edge becomes insulating, and hence trivial, with zero BDI invariant $\nu_{\mathrm{BDI}}^{\mathrm{ZZ}} = 0$ (yellow region). This phenomenon is shown in Fig. 5(a). In Fig. 5(b) we see that at finite $\mu$ we can cross from the $\nu_{\mathrm{BDI}}^{\mathrm{AC}} = 2$ regime (yellow region, metallic armchair) to the conventional D-class $\nu_D^{\mathrm{AC}} = \nu_{\mathrm{BDI}}^{\mathrm{AC}} = 1$ regime (salmon-colored region, spinless helical armchair), whereupon the number of MBSs per edge is halved, from two to one, following a band inversion. Degenerate BDI-class MBSs are expected to survive as near-zero modes

if the BDI-breaking effect of Rashba coupling $\lambda_R$ on the armchair edge states is finite but small, as is the case for typical experimental values of $\lambda_R \sim 1-5$ meV, see Fig. 5(f). In contrast, increasing $\lambda_R$ leaves the MBSs in the salmon-colored region completely unaffected in large enough samples, see Fig. 5(d), since in this case the D-class armchair edge remains topologically non-trivial.

## IV. EFFECT OF DISORDER AND MISALIGNMENT

Up to this point all our results have assumed perfect crystallographic armchair and zigzag edges. In real samples it is impossible to avoid a certain degree of misalignment when fabricating the superconducting contacts, or to create some amount of disorder. Since MBSs are topologically protected states, they should withstand such perturbations to a certain extent, but it is far from clear a priori if they are resilient to a realistic degree of misalignment and disorder. In this section we attempt to address this question by simulating the spectrum of a sample in configuration B with a fraction of vacancies along each edge and a finite rotation of the lattice.

Figure 6 compares the spatial localization of BDI-class and D-class MBSs on pristine, unrotated samples (a, b) and in samples with a 1% contact disorder and with a $2°$ contact misalignment (c, d). Disorder is introduced in our simulation in the form of vacancies at the given fraction of terminal sites along contact edges, removing any dangling bonds that are produced. While disorder and misalignment degrade MBS localization, for the device parameters considered they are found to remain spatially decoupled at this level of contact imperfections. Disorder above $\sim 2\%$ or misalignments above $7°$ leads to a splitting of MBSs due to edge leakage and overlap. We also quantitatively show in panels (e, f) the size of the minigap and degree of MBS splitting in contacts without disorder, purely as a function of the misalignment angle. We find that, at least in the regimes explored in our simulations, the BDI-class MBS minigap is actually more resilient to misalignment than the one of D-class MBSs. The latter tend to delocalize faster and to exhibit a minigap that becomes quickly polluted by low-lying states as the angle is increased. Above a $5°-7°$ misalignment, the MBSs in both cases are found to quickly merge into an edge-state quasicontinuum.

## V. JOSEPHSON EFFECT

A hallmark consequence of an odd number of MBSs at either side of a Josephson junction is the development of an anomalous $4\pi$-periodic Josephson effect[41,74] in the superconducting phase difference $\phi$ across the junction. A short (in $x$) and wide (in $y$) junction in configuration B, with a superconductor split as in Fig. 1(b), can be operated by tuning $\phi$ across the split weak link. As the junc-

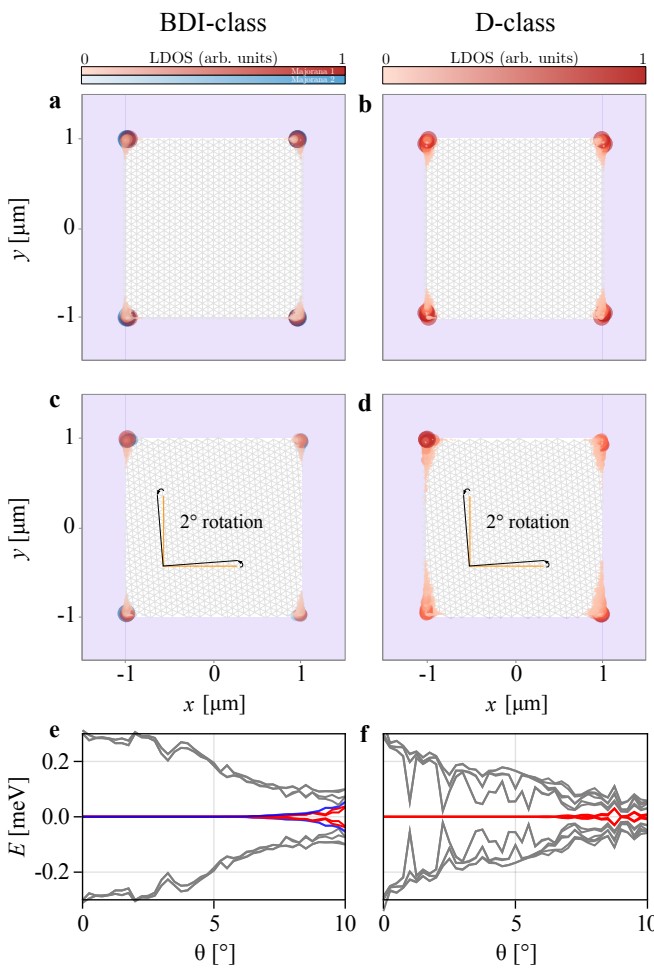

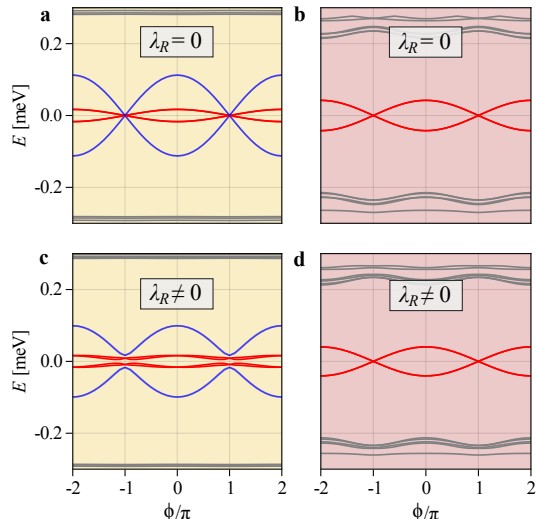

FIG. 7. Andreev levels as a function of superconducting phase bias in a Josephson junction similar to Fig. 1(b), for $\lambda_R = 0$ (a,b) and $\lambda_R = 2$ meV (c,d). The value of $E_Z$ is fixed to the vertical dashed line of Fig. 5(b) (a,c) and Fig. 5(c) (b,d). The width of the junction is $W = 2.5\mu m$, but the length is shortened to $L = 0.2\mu m$ to increase the phase-dependent MBS hybridization across the junction. Both the pairs of BDI-class MBSs in (a) and the lone D-class MBSs in (b) give rise to an approximate $4\pi$ Josephson effect. Increasing $\lambda_R$ breaks the BDI symmetry of (a), making the Josephson effect from corner Majorana pairs (near-zero modes) $2\pi$-periodic, and hence trivial (c). The $4\pi$ case of a single MBS per corner is however unaffected by Rashba (d).

FIG. 6. Spatial profile of MBSs, both in pristine (a,b) and imperfect (c,d) samples in configuration B. Left and right columns correspond, respectively, to BDI-class and D-class MBSs [for parameters marked with dashed lines of Figs. 5(b) and 5(c)]. In (c,d) contact disorder is 1% and misalignment angle is 2°. (e,f) Misalignment angle dependence of the low-energy spectrum in otherwise clean samples. In the BDI-class we depict the two Majoranas at each corner in blue and red, while the lone MBSs in the D-class are shown in red. The scaling of the lattice constant in the simulation has only a small effect on the magnitude and evolution of the topological gap in (e,f).

tion is assumed much wider than the size of the MBSs, it should behave as two Josephson junctions in parallel (one along each zigzag edge).[75] In the D-class ($\lambda_R \neq 0$) the device can be tuned to host either one or zero Majoranas per corner, which should produce an Andreev spectrum and Josephson current with $4\pi$- or $2\pi$-periodicity in $\phi$, respectively. In the BDI-class ($\lambda_R = 0$), Majorana pairs at a given corner will be decoupled from each other, so they should produce a $4\pi$-periodic spectrum and super-current.

We confirm these expectations, first for $\lambda_R = 0$, both for $\nu_{\text{BDI}}^{\text{AC}} = 2$ MBSs per corner, Fig. 7(a), and $\nu_{\text{BDI}}^{\text{AC}} = \nu_{\text{D}}^{\text{AC}} = 1$ MBS per corner, Fig. 7(b). Breaking BDI

symmetry with a finite $\lambda_R = 2$ meV makes the invariant trivial, $\nu_{\text{D}}^{\text{AC}} = 0$, so the Josepshon effect becomes $2\pi$-periodic, Fig. 7(c). Again, D-class Majoranas are unaffected by Rashba, and remain $4\pi$-periodic, Fig. 7(d).

## VI. CONCLUSION

We have shown that bilayer graphene, proximitized by laterally contacted superconductors and vertically encapsulated in transition metal dichalcogenides, exhibits a phase diagram with several topological phases below the spin-orbit bulk gap induced by the encapsulation. It includes non-trivial phases with single or pairs of MBSs at each armchair/zigzag corners, depending on the induced Rashba coupling. The system's phase can be controlled by tuning the chemical potential and an in-plane Zeeman field in ranges of the order of the bulk spin-orbit gap and the induced superconducting gap, respectively. The mechanism behind the topological phases is directly connected to the distinct properties of armchair and zigzag edges and the type of boundary modes they develop as a result of the SOC induced by the encapsulation. Despite the requirements of concrete arcmhair/zigzag crystallographic edges, a finite tolerance of around $\sim 5°$ in contact misalignment and $\sim 1\%$ in contact disorder is predicted,

making experimental realizations feasible.

A brief comparison of the above proposal to Majorana nanowires, as the current leading platform for MBSs, is in order. The two approaches exhibit interesting differences. One disadvantage of graphene is the g-factor, which is smaller ($\sim 2$) than in semiconducting nanowires ($\sim 2-18$, depending on details such as degree of metallization). As both approaches require a Zeeman energy comparable or greater than the induced superconducting gap, this demands stronger magnetic fields for comparable induced gaps. It has been shown, however, that highly controllable contacts and induced gaps are possible in graphene by using robust Type-II superconductors such as MoRe, that have much larger critical fields than Aluminum (the superconductor of choice for Majorana nanowires). This makes graphene's reduced g-factor potentially less of an issue. The problem of disorder also exhibits very different characteristics. In nanowires, charged defects and other sources of disorder are considered one of the most important challenges towards realizing MBSs[16,17]. In graphene-based van der Waals heterostructures, a very good control of puddles and bulk disorder is now possible using particular 2D crystals as substrates, such as hBN and graphite[76,77]. It is also to be expected that potential disorder will scatter electrons very differently in graphene than in semiconductors. Finally, the scale of energies that trigger multimode physics in our proposal is $\lambda_I \sim 2-10$ meV, which is larger than in typical low-density nanowires ($\hbar^2/[2m^* R^2] \sim 0.5$ meV). All these differences suggest that encapsulated graphene is worth exploring as a potential alternative to Majorana nanowires.

## METHODS

All our tight-binding simulations were performed using Quantica.jl[79]. All the code is available at Zenodo[80].

## ACKNOWLEDGMENTS

We thank Srijit Goswami and Prasanna Rout for valuable discussions.

**Competing interests:** The Authors declare no Competing Financial or Non-Financial Interests.

**Funding:** This research was supported by the Spanish Ministry of Economy and Competitiveness through Grants FIS2017-84860-R, PCI2018-093026 (FlagERA Topograph), PGC2018-097018-B-I00 and PID2021-122769NB-I00 (AEI/FEDER, EU), and the Comunidad de Madrid through Grant S2018/NMT-4511 (NMAT2D-CM).

**Author contributions:** F. P. prepared the numerical codes and performed the numerical simulations. F. P. and P.S-J. processed the data and prepared the figures. P.S-J. designed and oversaw the project. All authors contributed to discussing the physics and writing the manuscript.

## Appendix A: Resilience of helical modes in the presence of arbitrary edge orientation and scalar disorder

In the main text we have analyzed the emergence of helical edge modes in the armchair and zigzag edges by analyzing their respective bandstructures. Their robustness against disorder and edge misalignment is also studied, but only at the level of the p-wave phase and the associated MBSs in an SC-N contact. This does not directly address, however, the question about the general stability of the original helical states even before proximitization with a superconductor. We have argued that the induced SOC does not open a true topological-insulator gap, whose edge modes would be protected against any time-reversal-invariant perturbation by virtue of the bulk topology. Instead, the $2\pi$ Berry phase of the bilayer spectrum makes the SOC gap topologically fragile, meaning that time-reversal-symmetric perturbations such as Rashba may in principle destroy the associated pairs of helical edge states. In this appendix we demonstrate that this is not the case, at least for conventional spin-independent graphene imperfections, such as invervalley scattering at the edges or charge puddles.

As discussed in the main text, counterpropagating edge states have exactly opposite out-of-plane spin $s_z$ in both the armchair and zigzag cases. This is true even if we include realistic Rashba couplings, to a good approximation, due to its subleading contribution in the low-energy sector. Hence, backscattering of the edge states requires a spin-flip for both crystallographic orientations. Conventional scalar disoder should then be unable to localize edge states. We now show that this is also true for any other edge orientation, even in the presence of disorder.

Figure 8(a) shows the total density of states (DOS) in a sample of encapsulated bilayer graphene, with zero Zeeman and Rashba, and shaped like a circular strip. It is computed using the Kernel Polynomial Method[81]. The circular geometry has edges that vary across all possible crystallographic orientations. Figure 8(b) shows the corresponding spin-resolved current densities $\vec{J}_{\uparrow,\downarrow}$ in real space for all states within an energy window around neutrality (yellow box). The states were obtained by exact diagonalization using the Arnoldi method.

We see that despite the varying edge orientation around the circular strip, the DOS remains finite (ungapped) throughout the SOC gap (here from -10meV to 10meV). All these subgap states are spatially localized at the boundaries of the sample, and carry a net spin current, just as in the armchair and zigzag cases. This shows that edge states remain robust and gapless in the presence of arbitrary variations of edge orientation. The analysis can be extended by adding disorder. We apply strong Anderson disorder througout the whole circular strip, uniformly distributed in the interval

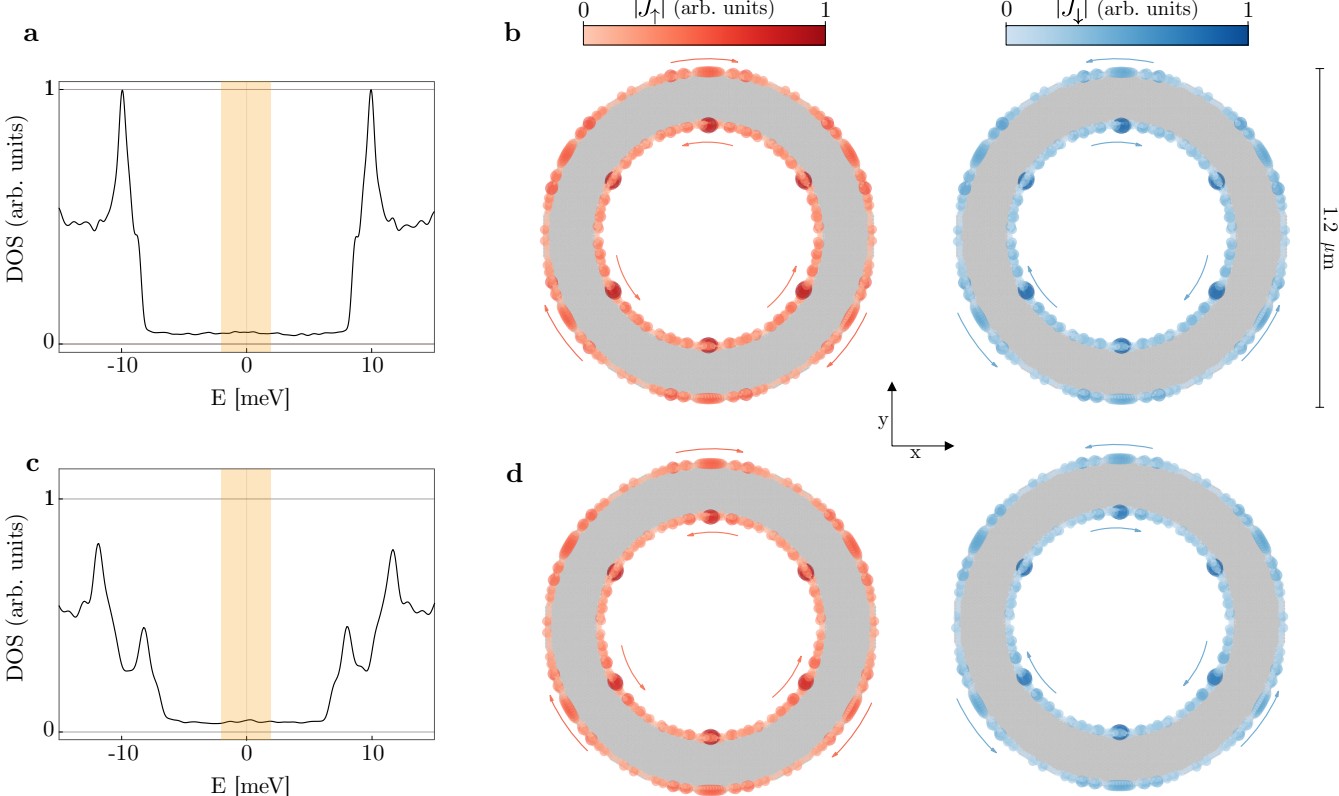

FIG. 8. (a) Total density of states (DOS) of encapsulated bilayer graphene without Rashba or Zeeman couplings, with $\lambda_I = 20$meV, and with the shape of a circular strip. The subgap DOS comes from edge states that remain ungapped all along the boubdaries of the sample, which span all crystallographic orientations from armchair to zigzag. The edge states carry a spin current, shown in (b) for states within the yellow strip. The current is not suppressed by the varying edge orientation, since localization would require spin-flip-backscattering. Curved arrows indicate current direction, and colored circles encode current magnitude. (c,d) The same as in (a,b) but with the addition of Anderson disorder of amplitude 2meV and zero average, distributed throughout the sample. Again, the spin-independent nature of the disorder leaves the edge states unperturbed, even though the DOS around and above the gap edge is strongly affected.

$[-2\text{meV}, 2\text{meV}]$. The result is presented in Fig. 8(c,d). While disorder has a strong effect on the DOS outside the gap, the subgap DOS remains unperturbed. Likewise, the edge current remains insensitive to the disorder.

We thus find that the helical edge states behave as true topological modes protected against arbitrary perturbations, as long as they are spin-independent, or at most commute with $s_z$.

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
