# Peer review of "Majorana bound states in encapsulated bilayer graphene"

_SciPost Physics_

## Round 3 · Referee Report · Antonio Manesco (Referee 1) · 2022-7-14

Strengths

1- The authors introduce a new way to achieve topological superconductivity in a graphene-based device that does not require electron-electron interactions or strong magnetic fields.

Weaknesses

1- The proposed system requires fine-tuning of the lattice orientation. 2- The effects of the parenting superconductor are not treated on the same footing as the graphene region. It could be that the robustness of the device is overestimated.

Report

The authors introduce bilayer graphene encapsulated with transition metal dichalcogenides as a platform for Majorana bound states. Different from the previous proposals, this method does not require electronic interactions and/or large magnetic fields to create the helical channels necessary to obtain p-wave pairing. Instead, the helical edge states result from the proximitized spin-orbit coupling due to the encapsulation.

Although I understand the benefits of this new platform in comparison with the previous ones, I believe the proposed device is beyond what could be reached by the current state-of-art fabrication techniques. With small changes to the current calculations toward more realistic models, the authors could either show the robustness of their results or point out the limitations and challenges on the experimental side. I provide detailed questions and suggestions below.

Finally, this submission does not meet the criteria of Scipost Physics but does meet those of Scipost Physics Core, where it could be published after the listed queries are properly addressed.

1- The main results in the manuscript rely on having an armchair interface between graphene and the superconductor. Is there an understanding of why this is the case? What is different in a zigzag interface that does not allow the appearance of Majorana zero modes? For example, along the armchair direction, the valley number is ill-defined. Does it play a role?

2- The authors mention MoRe as a suitable parent superconductor. The current fabrication techniques of MoRe superconducting contacts likely introduce strong disorder in the superconducting region. Since the superconducting gap is rather small, I expect the wavefunctions to enter the superconducting region. Since the authors mentioned that they are capable to perform simulations of a square lattice superconductor, I wonder whether they still would observe Majorana zero modes if onsite disorder is added to the superconducting region. Does this result relate to the specific crystal orientation (armchair)?

3- I also expect the superconductor to introduce a sizable Fermi level mismatch at the normal/superconductor interface. Moreover, due to charge screening, I expect that the resulting electrostatic potential will be smooth over distances comparable with the lattice constant. How does a smooth potential at the normal/superconductor interface change the result?

4- The authors discuss that, in the absence of Rashba spin-orbit coupling, the system Hamiltonian belongs to the BDI symmetry class. The reasoning follows a comparison with multiband nanowires: when the bands are uncoupled, multiple Majorana modes coexist. Is this result robust in the presence of disorder/intervalley scattering? I believe that in this case the multiple bands would also be coupled by elastic scattering. Moreover, what is the ratio between the energy splitting of the end modes and the topological gap size in the presence of Rashba spin-orbit coupling?

Requested changes

1- Add simulations with a disordered superconductor.

2- Provide an intuition on why an armchair interface is needed.

3- Add a Fermi level mismatch between graphene and the superconductor.

4- Check whether BDI class survives in the presence of elastic scattering.

5- Compare energy splitting with Rashba and topological gap when there are two coexisting zero modes at each end.

  • validity: good
  • significance: ok
  • originality: good
  • clarity: high
  • formatting: excellent
  • grammar: excellent

Author:  Pablo San-Jose  on 2022-11-25  [id 3077]

(in reply to Report 1 by Antonio Manesco on 2022-07-14)

Dear Referee,

we thank you for your time and your detailed report. Below we comment on your criticisms.

Regarding your four bullet points:

1- The need to couple the graphene bilayer to a superconductor is the same as in the case of hybrid nanowires. The standard Fu-Kane recipe for topological superconductivity and Majorana bound states is to couple a spinless helical mode to an s-wave superconductor. In the case of a zigzag edge it is not possible to realize a spinless helical mode to start with, so coupling it to a conventional superconductor just opens a trivial s-wave gap, not a topological p-wave gap as in the armchair edge. Valley conservation does not play an important role in this argument.

2- While it is possible to compute the electronic structure with a superconducting square lattice instead of a self-energy, both calculations become equivalent for large enough superconductors. However, the self-energy computation is much, much more efficient, numerically. The addition of disorder can be done either on the graphene side or on the superconductor side. We do the former in this work, but we expect both to yield similar results. Considering disorder only on the superconducting side could potentially be done using the Usadel approach to diffusive superconductivity. Doing this would likely underestimate the destructive effect of disorder on the topological gap respect to the approach in our manuscript. The reason is that in the diffusive limit we loose all inhomogeneities in the superconducting self-energy (it becomes spatially uniform), which would therefore suppress all disorder-induced scattering processes. It would merely renormalize the pairing amplitude on graphene.

3- We have briefly looked at the effect of inhomogeneous charge distributions at the contact, and the associated multimode edges. The results are qualitatively the same. Charge accumulation pushes more propagating modes into the contact, but the total number is still odd in the spinless helical window. Hence, the effect of the proximity effect is still the same: a p-wave gap is opened (of a magnitude that depends on the details of the superconductor-edge mode coupling and hence on the charge accumulation profile), and MBSs appear. The detailed shape of the phase diagram, however, is expected to depend on the charge accumulation, as it shifts the energy of the bands. We comment on these multimode issues at the end of the conclusion.

4- Figure 6(c,e) shows precisely how resilient BDI-class MBSs are respect to the addition of vacancies and to angle misalignments. Somewhat surprisingly, the BDI-class MBSs appear to be slightly more resilient to imperfections than D-class MBSs in our simulations. Regarding the amount of splitting of BDI-class MBSs in the presence of Rashba coupling, this is shown in Fig 5d: the splitting is much smaller than typical gaps for Rashba SOCs below 10meV.

For your convenience, please find a PDF of the resubmitted manuscript with all changes marked in red at the code repository: https://github.com/fernandopenaranda/MBSinBLG/blob/main/manuscript/encapsulated.pdf

---

## Round 3 · Referee Report · Anonymous (Referee 2) · 2022-10-29

Strengths

1- The manuscript is well-written and the presentation is clear. 2- Figures are nearly all clear and effective. 3- Code and data are openly available. 4- May provide a new platform for the realization of Majorana fermions.

Weaknesses

1- Ideas are now standard in the literature. 2- Challenges associated with old proposals seem to be present for the ideas proposed here. 3- Arguments for robustness of Majorana fermions are not strong.

Report

The authors study a Bernal-stacked graphene bilayer encapsulated in transition-metal dichalcogenides (such as WSe${}_2$ and WS${}_2$), in which Rashba and Ising spin-orbit couplings are significantly enhanced compared to native single- or bi-layer graphene. They argue that the spin-orbit coupling produces topological insulators with edge states along armchair and zigzag directions with different spin structures. This structure can in turn be used to produce a helical 1d edge state along the armchair edge in the presence of an in-plane Zeeman field.

Finally, the authors study the proximity induced superconductivity of these edge states and find, due to the helical nature of the armchair states, that they can support Majorana fermions at the corners of the sample where armchair and zigzag edges meet. Depending on the strength of the Zeeman and Rashba terms, these Majorana fermions may be characterized by a $\mathbb{Z}_2$ or $\mathbb{Z}$ invariant int the D or the BDI symmetry class, respectively, and exhibit $4\pi$ -periodic energy spectra in a Josephson junction geometry.

The results are presented carefully and supported by tight-binding numerical simulations, which are also available openly on Zenodo. They offer a new possible platform to realize unpaired Majorana fermions, which could address some of the challenges with previous proposals. However, this referee does not see any new insights or ground-breaking advance in ideas that would avoid problems that have plagued experimental realization of Majorana fermions based on previous proposals such as disorder, sensitivity to magnetic fields, etc.

Overall, I believe this work is interesting and satisfies the criteria for publication in SciPost Physics after some changes.

Requested changes

1- Provide reasoning or clear evidence that the spin of armchair edge states is out of plane. Make this clear, especially under Eq. (4). 2- Provide reasoning or clear evidence for the different spin structures of the armchair and zigzag edge states. 3- Provide a legend / color bar for the density of states shown in Fig. 4(c) and 4(d) and, similarly, for Fig. 6(a), 6(b), 6(c), 6(d).

  • validity: good
  • significance: ok
  • originality: ok
  • clarity: high
  • formatting: excellent
  • grammar: excellent

Author:  Pablo San-Jose  on 2022-11-25  [id 3076]

(in reply to Report 2 on 2022-10-29)

Dear Referee,

we thank you for your detailed report and for the recommendation to publish after some changes.

We have addressed each of the required modifications: - We have clarified the origin of the out-of-plane spin polarization of edge modes after Eq. (4), which is due to the Ising spin-orbit coupling. - This question is probably due to confusion caused by a wrong assignment of spins in Fig 2. It was indeed wrong and has been corrected (see also response to Prof. Wimmer's report). Moreover, it is now more explicitly stated that the spin structure of all edge modes follows the standard spin-momentum locking expected of spin-orbit coupling. - We have added the color bars as requested.

For your convenience, please find a PDF of the resubmitted manuscript with all changes marked in red at the code repository: https://github.com/fernandopenaranda/MBSinBLG/blob/main/manuscript/encapsulated.pdf

---

## Round 3 · Referee Report · Michael Wimmer (Referee 3) · 2022-11-5

Strengths

  • Experimentally relevant proposal
  • material choice may have advantage over existing Majorana proposal
  • Code avalaible on zenodo

Weaknesses

  • some key aspects could be explained better in the manuscript
  • BDI-invariant is not clear

Report

The current manuscript deals with engineering Majorana bound states in encapsulated bilayer graphene. The past years have shown that existing experiments towards Majorana bound states are most likely handicapped by material problems and impurities, and as such proposals that can overcome this problem are very much welcome. The proposal of this manuscript is therefore based on using a variant of encapsulated graphene that is known for very low disorder and ballistic transport. As such, I find this proposal very interesting. As a potential breakthrough of a long-standing problem, it would also fit in my opinion with SciPost Physics.

I however do have a number of questions/suggestions to the authors that should be addressed before publication:

1.) In general, I found it hard to follow the discussion of gap opening after Eq. (4), and in particular I was rather confused by Fig. 2. I first thought the color indicates the spin direction. In particular, the sketch of edge states seemed to indicate for the zigzag edge that the color also corresponeded to a particular spin-direction (positive or negative z-direction). But I then realized that the sketch must be true only for some energy - spins cannot simply flip across a band crossing.
In fact, I am wondering if indicating the spin-direction by color instead wouldn't be a better presentation, as it seems that this is what is needed to understand where gaps open. Or is the current color information needed to understand the physics?

2.) While I trust the numerical results regarding the nearly degenerate states from section III, I am unsure about the argumentation with regards to BDI. In the absence of Rashba spin-orbit coupling, spin in z-direction is a good quantum number. Hence, it seems to me that it would be more natural to argue from the point of view of the two spin-subblocks of the Hamiltonian instead, and I suspect that each of them independently hosts a zero-energy state. This is different though from a true class BDI system, where the pesence of a TRS squaring to 1 is essential. For example, the authors state that for the armchair edge the BDI-invariant is 2 at E_Z=0. However, in this case there surely is also the TRS squaring to -1, hence classifying this case as BDI seems not correct.
This should be clarified and stated more precisely. If I am wrong, and there is indeed a proper BDI invariant/symmetry, it should be explicitly given.

3.) In Section IV, the authors study the stability of the Majorana bound states with regards to misalignment. I was wondering if the results for the maximally allowed angle could be influenced by the fact that the authors used a scaled lattice constant? Specifically, it is known that for a large enough distance in terms of lattice constant, any non-armchair edge is equivalent to a zigzag edge (Phys. Rev. B 77, 085423 (2008)).
I believe it would be useful to discuss this aspect, or possibly show that the maximally allowed misalignment does not depend on the value of the lattice constant/size of the system.
  • validity: ok
  • significance: high
  • originality: high
  • clarity: ok
  • formatting: good
  • grammar: good

Author:  Pablo San-Jose  on 2022-11-25  [id 3075]

(in reply to Report 3 by Michael Wimmer on 2022-11-05)

Dear Prof. Wimmer,

thank you for your detailed report and for the positive recommendation to publish. We answer your three questions below:

1) The explanation of the splitting was in fact not only unclear but actually wrong for the very reasons you mention: we had misidentified the spins in the old Fig 2, which actually led to your confusion regarding the spin of crossing bands. We had also failed to explain clearly how each of the crossings became split.

We have now thoroughly revised Fig 2 and its discussion. We have now corrected the spin labeling of all edge modes, and have changed the red/blue color code to denote spin as you suggest. As it turns out, the spin structure of edge modes is always helical in pairs, both for zigzag and armchair.

The correct spin structure of the subgap bands raises the question of why some of the crossings don't get split by an in-plane Zeeman. In particular, the finite energy crossings at the M point for zigzag, or the zero energy crossings at finite k_y for armchair (when Rashba is zero) remain unsplit. Why? We encountered this behavior initially as a numerical fact, but we have now provided an explanation in terms of additional orbital symmetries of the crossing states. These symmetries work differently for armchair and for zigzag. For the armchair case the unsplit crossings can actually be classified into opposite sectors of the parity operator, so it is a crystal symmetry which is protecting that crossing. In the case of the zigzag edge modes, an analogous symmetry exists, although only at precisely the M point where the unsplit edge mode cross. This is now explained in detail before and around Eq. (4).

We have also improved the labeling of Fig 2 to illustrate the meaning of the four phases (white, purple, salmon, yellow) in terms of the band structure, which will hopefully help the reader interpret the rest of the results more easily.

2) You make a very good point regarding the BDI classification. At E_Z=0 there are two symmetries that get compounded, the actual time-reversal symmetry T (TRS), which squares to T^2=-1, and the pseudo-TRS of simple conjugation K, which squares to K^2 = +1. The latter is the one introduced in Ref. 64 (Tewari & Sau '12) as the mechanism behind the hidden BDI symmetry of narrow nanowires. The Z invariant of BDI is strictly applicable, in analogy to that paper, when T is broken (i.e. at E_Z>0), so that the Hamiltonian is no longer simply the direct sum of two decoupled spin sectors. This was a mistake in the original manuscript. We have now changed our phrasing to make the symmetry analysis more precise.

3) We have checked that our simulations of the topological gap versus edge angle remain qualitatively unchanged when varying the lattice constant scaling. The precise scaling does affect the existence and position of isolated subgap states in some instances (see lowest states above zero in Fig. 6f), which are therefore probably difficult to predict without ab-initio calculations. The magnitude of the gap, and importantly also the degree of MBS splitting do not depend significantly on the scaling. Please see the following figure with a comparison of different scalings. https://github.com/fernandopenaranda/MBSinBLG/blob/main/data/figdisordervsscaling/disordervsscaling.pdf

Regarding the universal zigzag-like behavior of arbitrary edges you mention, it is our understanding that those results apply to the far-field scattering amplitudes of delocalized modes impinging on arbitrary chiral edges. In our case we are concerned with evanescent modes localized at the edge, so we are not sure how those results translate to our work. What is clear from our numerical simulations is that the p-wave gap survives small variations away from the armchair orientation. Perhaps there is an interplay between induced p-wave gap and critical contact misalignment, but we were not able to find a precise connection numerically.

For your convenience, please find a PDF of the resubmitted manuscript with all changes marked in red at the code repository: https://github.com/fernandopenaranda/MBSinBLG/blob/main/manuscript/encapsulated.pdf

---

## Editorial Decision

resubmitted